# Challenges in the distribution of antimicrobial medications in community dispensaries in Accra, Ghana

Hannah Camille Greene[1], Kinga Makovi[1], Rafiatu Abdul-Mumin[2], Akhil Bansal[3], Jemima A. Frimpong[1]*

**1** Social Science Division, New York University Abu Dhabi, Abu Dhabi, United Arab Emirates, **2** Department of Biochemistry and Forensic Sciences, School of Chemical and Biochemical Sciences, C.K Tedam University of Technology & Applied Sciences, Navrongo, Ghana, **3** Faculty of Medicine and Health, University of Sydney, Sydney, Australia

* jafrimpong@nyu.edu

**Data Availability Statement:** The data underlying the results presented in the study are available at the following URL: https://www.openicpsr.org/openicpsr/project/192062/version/V2/view.

## Abstract

### Introduction

The dispensation of medicines in some low- and middle-income countries is often carried out by private vendors operating under constrained conditions. The aim of this study was to understand the challenges reported by employees of dispensaries, specifically, chemical and herbal shops and pharmacies in Accra, Ghana. Our objectives were twofold: (1) to assess challenges faced by medicine vendors related to dispensing antimicrobials (antibiotic and antimalarial medications), and (2) to identify opportunities for improving their stewardship of antimicrobials.

### Methods

Data were collected in 79 dispensaries throughout Accra, in 2021, using a survey questionnaire. We used open-ended questions, grounded on an adapted socioecological model of public health, to analyze these data and determine challenges faced by respondents.

### Results

We identified multiple, interlocking challenges faced by medicine vendors. Many of these relate to challenges of antimicrobial stewardship (following evidence-based practices when dispensing medicines). Overall, medicine vendors frequently reported challenges at the Customer and Community levels. These included strained interactions with customers and the prohibitive costs of medications. The consequences of these challenges reverberated and manifested through all levels of the socioecological model of public health (Entity, Customer, Community, Global).

### Discussion

The safe and effective distribution of medications was truncated by strained interactions, often related to the cost of medicines and gaps in knowledge. While addressing these

**Funding:** H.G. acknowledges funding for the fieldwork by the New York University Abu Dhabi Divi- sion of Social Science, as well as a grant from the Effective Altruism Infrastructure Fund (a3r6g000001Tac2AAC). The funders had no role in study design, data collection and analysis, decision to publish, or preparation of the manuscript. K.M. acknowledges research funds by the NYUAD Center for Interacting Urban Networks (CITIES), funded by Tamkeen under the NYUAD Research Institute Award CG001. The funders had no role in study design, data collection and analysis, decision to publish, or preparation of the manuscript.

**Competing interests:** The authors have declared that no competing interests exist.

challenges requires multifaceted approaches, we identified several areas that, if intervened upon, could unlock the great potential of antimicrobal stewardship. The effective and efficient implementation of key interventions could facilitate efforts spearheaded by medicine vendors and leverage the benefits of their role as health educators and service providers.

## Conclusion

Addressing barriers faced by medicine vendors would provide an opportunity to significantly improve the provision of medications, and ultimately population health. Such efforts will likely expand access to populations who may otherwise be unable to access medications and treatment in formal institutions of care such as hospitals. Our findings also highlight the broad range of care provided by shopkeepers and vendors at dispensaries. These findings suggest that the meaningful engagement of dispensaries as valued conduits of community health is a promising pathway for interventions aiming to improve antimicrobial stewardship.

## Introduction

Antibiotics and antimalarials are critically important life-saving medications. However, their misuse and overuse can accelerate the emergence of antimicrobial resistance (AMR) [1–3]. AMR is projected to cause 10 million deaths annually by 2050 [4]. Major drivers of AMR outside the scope of hospitals include overdistribution and inappropriate use [5, 6]. Distributing antimicrobials at high rates increases the likelihood of their misuse, often through imprudent consumption for abiotic infections, or low-strength doses. Another key contributor of misuse includes consumer failure to adhere to entire treatment courses [7], which increases the risk of evolving drug-resistant disease outbreaks [8]. When incomplete or substandard treatment courses are used, resilient microbes with genetic mutations conferring resistance to an active ingredient may evolve [9, 10]. In such cases, the original medication can become ineffective, with fatal consequences [11, 12]. Resistant pathogens spread particularly rapidly in communities without a preventive public health infrastructure, such as reliable clean drinking water, which exacerbates their impact [1].

Pharmacies and chemical shops in low- and middle-income countries play a key role in how medications and medical advice are accessed. This is especially so in the absence of formal institutions of healthcare that have established practices of restricting or adjusting the use of antibiotics through diagnostic testing and clinical observation. Employees in pharmacies and chemical shops therefore are key gatekeepers of medications [13]. However, the best practices for stewardship (following evidence-based practices in dispensing antimicrobials, including education, dosage, duration, etc., in an effort to reduce AMR and improve patient outcomes) at the community level, where these entities operate, are less clear [14]. Challenges faced by employees of chemical shops and pharmacies that might prevent them from practicing the judicious distribution of antimicrobials are not fully understood, as only a few studies have focused on these factors [15–17].

Ghana presents an opportune context for studying these challenges in the distribution of antimicrobial medications in community settings. First, Ghana has one of the highest burdens of malaria, which persists as a perennial and endemic disease [18]. Not surprisingly, drug-resistant malaria and bacterial pathogens have also been identified in Ghana [19, 20]. Second, despite progress toward the elimination of some pathogens, Ghana still faces endemic diseases

that require antibiotics for treatment [21]. The presence of these diseases is compounded by access to medical institutions, which is often impeded by cost that is prohibitive for populations with sporadic and low wages [22].

Health financing in Ghana is primarily based on out-of-pocket spending as a result of the limited uptake of the national health insurance scheme [23]. When the Ghanaian government-subsidized healthcare system was changed to a fee-based "cash-and-carry" system in 1985 [24], pharmacies and chemical shops quickly emerged as key entities of the health-seeking/health-provision infrastructure. In this context, pharmacies are licensed to sell a variety of medications, some of which require prescriptions or can only be distributed at the discretion of a licensed pharmacist [23]. Over-the-counter medicine sellers, widely referred to as chemical shops, are licensed only for the distribution of non-prescription drugs [25, 26]. Prior literature found frequent self-medication through private medical shops in Ghana and beyond [25, 27]. This prior work pointed to pharmacies and over-the-counter medicine shops alike as sites of unrestricted, ill-informed antibiotic overdispensation and for this reason, disproportionate drivers of AMR [28–31].

As showcased by these examples, previous studies have addressed pharmacies and chemical shops as entities to be studied, with publications written *about* vendors rather than *with* their input. Thus, this prior work fails to highlight the complex experiences of those working within community health contexts (as in most studies addressed in reviews [32, 33]). The aim of the current study is therefore to assess the context-specific perspectives of medicine distributors in Accra, Ghana, focusing on their sales of antibiotic and antimalarial medications (for prior examples, see [34–36]). Recognizing the rich potential of medicine vendors to mitigate outbreaks of preventable and treatable diseases, we sought to understand the barriers faced in their provision of medications to communities, from their perspective. We aimed to elucidate systemic challenges that transcend health outcomes beyond the walls of medicine shops. Our findings contribute to better care and stewardship of antibiotics and antimalarials at the community level, for improved service delivery and ultimately patient outcomes.

## Methods

### Study setting and context

Data were collected from November to December 2021 in 17 neighborhoods in the Greater Accra Region. Accra is densely populated and growing each year, as a result of regional migration and rapid urbanization [37]. A large proportion of Ghana's hospitals are located in Accra. Within the city, these hospitals are predominantly concentrated in wealthier neighborhoods, a result of colonial segregation and its persistent ramifications in Ghana's development [38]. By 2022 a network of approximately 4,198 registered pharmacies and 20,326 licensed over-the-counter medicine sellers has emerged throughout Ghana [39]. The number of each type of dispensary in Accra is not reported, and very few shops have a web presence or location documented on maps. The Ghana Pharmacy Council standardizes prerequisite courses for employees of medicine dispensaries, and requires that pharmacists complete at least four years of study before becoming a licensed Doctor of Pharmacy. Registered medicine counter assistants (MCAs) complete a six-month course before taking the Ghana Medicine Counter Assistants Certificate Examination. Pharmacy Technicians require the Ghana Pharmacy Technicians Qualifying Exam.

### Survey

We first performed a comprehensive review of the literature in combination with a series of discussions with several local experts. Our findings from the review and discussions informed

the design of the survey, which focused on barriers and facilitators to the dispensation of medicines. The questionnaire was piloted at five field sites and was revised on the basis of feedback received. The survey consisted of 72 questions and took an average of 45 minutes to complete. The complete questionnaire assessed the demographic background of the shopkeepers, sales traffic, prices, options for antimicrobial brands, common symptoms observed, and characteristics of the neighborhood surrounding each shop (see S1 File for the complete instrument). The surveys were administered verbally and simultaneously transcribed on a laptop or written in printed copies and later transcribed. The present paper relies primarily on responses to qualitative, open-ended questions, with a particular focus on the final question of the survey: "What is the biggest challenge you face as a medicine seller?".

## Inclusion and exclusion criteria

Any employee over 18 years of age in an herbal shop, chemical shop, or pharmacy in the Greater Accra Region who sold antibiotic and/or antimalarial medications was eligible for the survey. Surveys were completed in English or Twi. In total, 88 shops were visited, and 93% (n = 82) agreed to take part in the study. Potential respondents were selected based on a number of criteria, namely the availability of antimalarial or antibiotic medications at the shop, and the presence of an employee with the necessary knowledge and willingness to participate in the survey. Of the respondents who offered to participate, 79 met the inclusion criteria and completed the survey. Forty-four of the participating field sites were pharmacies, 32 were chemical shops, and three were herbal shops.

## Study sample

Data were collected in pharmacies and chemical or herbal shops. We intentionally sought out a diverse set of neighborhoods in various segments of the city. Distance from a hospital was also a key factor in our selection. Of the medicine dispensaries that were visible and approached, 93% agreed to participate. Two disproportionately lower-income neighborhoods, Nima and Maamobi, which are often neglected, including from research, were oversampled. Respondents were invited to participate by explaining the purpose of the project, describing the consent process, and offering a printed copy of the survey for review before deciding whether to participate. When multiple employees were present, one who met the criteria and was able to participate was collectively nominated to complete the survey. Participants were compensated 1.00 Ghanaian cedi per minute required by taking the survey (which corresponds to USD10.00 per hour).

## Ethical considerations

The study was approved by the New York University Abu Dhabi Institutional Review Board (HRPP-2021-146). A detailed consent form was distributed to participants after introducing the project verbally. Respondents were invited to decline to answer any question and withdraw at any time if they saw fit. Some surveys were administered on paper, others on a laptop using Qualtrics. The Qualtrics questionnaire automatically skipped questions about antibiotics for chemical shops, to investigate only the over-the-counter medicines they were legally licensed to sell. While this presents a limitation, it was essential to minimize participation risks for the respondents. No identifying information about shops or participants was recorded, and location data was kept only at the neighborhood level. For confidentiality purposes, surveys were conducted at a location directed by the participant.

## Conceptual framework

The study was informed by an adapted socioecological model of public health [40]. The model categorizes factors that influence health into multiple levels: individual, interpersonal, organizational, community, and public policy [40]. The unadapted model, centering patients, posits that examining these factors and their interactions could be essential for developing interventions and strategies to improve health and well-being. Similar to other studies, we adapted the model to better align with our study setting and context, i.e., systemic challenges and root causes that diverge from the original conceptualization of the model. For example, Golden and colleagues inverse the model, placing policies and environments as the first sphere in which communities, organizations, interpersonal connections, and then individuals are embedded [41]. The CDC distills the model into four categories rather than five: individual, relationship, community, and society [42].

In our adapted model, we center the perspective of medicine distributors, instead of patients. This was necessary given our focus on the supply side of medicine access. Along these lines, we use four instead of five categories to make the framework more applicable, enhance its analytical clarity, and create non-overlapping categories. We assign each challenge to the following categories: Entity, Customer, Community, Global. Entity refers to the medicine shop or pharmacy, and the challenges arising in shops, for shops, and those working in these entities. The next category is the Customer, and challenges arising from interactions with individual customers. Customers are then embedded in the surrounding community, hence the category, Community. Challenges in this category are identified when respondents referred to interaction patterns that arose from within the surrounding community, or interacting with "typical" customers that they found challenging. Finally, some challenges have clear global origins that point beyond the immediate community that medicine sellers serve, which are placed in the Global category. Detailed information on the categories, definitions, and examples can be found in Table S1 in the S1 File.

## Analysis

Responses to the question "What is the biggest challenge you face as a medicine seller?" span the various categories of the socioecological model of public health [40]. Responses were coded by one of the authors using a qualitative content analysis method: first coded into sub-themes and then into larger categories of the modified socioecological model. A second author independently carried out the same tasks, and disagreements were resolved in conversation. Where responses span multiple categories, i.e., when respondents mention multiple challenges all were coded separately. This ensured a comprehensive approach to distilling challenges, rather than speculating on which may have been the most important. We then analyzed the data for insights relevant to the categories of our conceptual framework.

Based on prior studies, challenges at the Entity level might include low job satisfaction [43], feelings of inadequate preparedness, time scarcity [17], and gaps in knowledge or ability to implement desired behaviors [16, 44]. For the Entity category, we created the following sub-codes: competition, low sales volume, and other workplace-related issues. Challenges in the Customer category were those occurring during the interactions between distributors and their customers. These include the following subcodes: customer attitudes towards distributors, adherence to medicines, which include various deviations from the prescribed or suggested regimens, attempts at self-medication [17, 45], and related issues that arise during the course of treatment. The Community category included challenges that manifest clearly through shopkeepers' interactions with typical patients in their communities. A way to distinguish between these and those at the customer level is to pay keen attention to the attributions

that distributors make. Challenges falling in the Community category are ascribed to more than one individual customer, and are identified as recurring patterns generalized to the community in which the shop is located. Based on prior work these could include limited geographic access to care [46] or diagnostic facilities [7], local socioeconomic inequities [47], and surrounding environmental challenges [6]. In our study, we find evidence of many of these and in addition, identify language barriers that put distributors at odds with their customers. The Global category encompasses challenges that originate beyond the community, driven by the broader political and economic circumstances affecting medicine distribution. In our study, these include supply chain issues and navigating discussions about medicine brands with a variety of implications.

## Results

The demographic composition of the sample is described in Table 1 by shop type. The majority of respondents were 34 years or under and 67% of the respondents were female (disproportionately more so in herbal or chemical shops). In terms of role, respondents included 17 pharmacists, 41 medicine counter assistants, 12 shop owners, and an "other employee" category. All but 11 respondents reported formal titles conferred by training programs. This corresponds to the self-reported information on the amount of training received by respondents. Specifically, pharmacists reported at least 4 years of training, and were concentrated in pharmacies, while those in herbal or chemical shops were more likely to report no training at all (about one in five of respondents reported no training overall in the sample).

Of those who reported only "shop owner" or "other employee," rather than licensed titles, five had completed secondary school, one had finished vocational training, and four had completed a Bachelor's degree. One reported that they had "*learned from my mom how to read prescriptions. She went to school. I employed a worker here and she went to school, and I learned it from her. I Google a lot and make research and learn a lot. Before the medicine comes, I learn a lot. I'm working with human beings so I have to. I learned more from the phone. I didn't go to school for medicine because my mind is exhausted. . . didn't want to waste money because I had not taken the form [exam]*" (female, 35–44 year old chemical shop owner in a self-ranked medium-income neighborhood). (Note that neighborhood income ranking is as reported by the respondent based on their own adjudication of the surrounding neighborhood.)

In terms of seniority, respondents had worked in their shops for a median of 2 years (mean = 5.0). In herbal or chemical shops, as well as pharmacies, one in four respondents worked at the location for less than a year. Respondents reported working a median of 8.5 hours per shift (mean = 9.8), six days per week, suggesting a typical 58-hour workweek. The workload appears substantially higher in herbal or chemical shops, where 97% reported working more than five days per week, compared to pharmacies, where 29% of respondents reported working only five days. Similarly, respondents in chemical and herbal shops reported working for about 4 hours longer each day, contributing significantly to their workload.

Table 2 shows the challenges reported, by shop type, and lists categories with their respective subcategories. The table is organized by the number of challenges mentioned by respondents in each shop type (44 in herbal and chemical shops, and 65 in pharmacies), rather than the number of respondents. In herbal and chemical shops respondents on average mentioned 1.3 challenges (85% of respondents mentioning only a single challenge), while the same statistic in pharmacies was 1.5 (61% of respondents mentioning only a single challenge).

10% of respondents reported that they do not perceive any challenges. It should be noted that being deeply embedded in the surrounding community led some of them to proclaim a lack of challenges, as several echoed the quote that they "*have not faced much challenge due to*

**Table 1. Demographic characteristics and work experience of respondents.**

|  | Herbal/Chemical Shop | Pharmacy |
|---|---|---|
| **N** | **35** | **44** |
| **Role** |  |  |
| Pharmacist | 0.00% | 38.64% |
| Shop owner | 31.43% | 2.27% |
| Medical Counter Assistant | 48.57% | 54.55% |
| Other employee | 20.00% | 4.55% |
| **Age** |  |  |
| 18–24 years old | 25.71% | 13.64% |
| 25–34 years old | 25.71% | 52.27% |
| 35–44 years old | 34.29% | 20.45% |
| 45–54 years old | 8.57% | 9.09% |
| 55–64 years old | 5.71% | 4.55% |
| **Gender** |  |  |
| Female | 71.43% | 63.64% |
| Male | 20.00% | 34.09% |
| Missing | 8.57% | 2.27% |
| **Education** |  |  |
| Senior high school or less | 68.57% | 47.73% |
| BA or higher | 17.14% | 50.00% |
| Other | 14.29% | 2.27% |
| **Training** |  |  |
| No training | 25.71% | 11.36% |
| 1–11 months | 54.29% | 47.74% |
| 12–23 months | 0.00% | 6.82% |
| 24 months or more | 20.00% | 34.09% |
| **Years working in shop** |  |  |
| Less than 1 | 22.86% | 29.55% |
| 1–3 | 20.00% | 36.36% |
| 4–6 | 20.00% | 11.36% |
| More than 6 | 37.14% | 22.73% |
| **Days of the week worked** |  |  |
| 5 | 2.86% | 30.23% |
| 6 | 54.29% | 41.86% |
| 7 | 42.86% | 27.91% |
| **# of hours worked/day mean (sd)** | 12.0 (3.7) | 8.2 (1.4) |

*familiarity*" (35–44 year old herbal shop owner in a low-income neighborhood). Such respondents may have also experienced the challenges listed by others, but did not perceive them as major hardships worth mentioning.

## Challenges at the Entity level

**Low sales and competition.** Medicine shops reported encountering fluctuations in sales, with seasonal variations being a key element of sales patterns. The use of medications varies seasonally, with demand for antimalarials peaking in the rainy season, while for other ailments peaking during the harmattan or dry season. For example, a respondent noted that sales of

**Table 2. Challenges reported by shop type.**

|  | Herbal/Chemical Shop | Pharmacy |
|---|---|---|
| N (number of challenges) | 44 | 65 |
| **Entity** | | |
| Low sales | 13.64% | 1.54% |
| Competition | 0.00% | 3.08% |
| Workplace | 0.00% | 3.08% |
| **Customer** | | |
| Attitude | 9.09% | 20.00% |
| Self-medication | 0.00% | 12.31% |
| Treatment | 9.09% | 3.08% |
| Adherence | 6.82% | 1.54% |
| **Community** | | |
| Payment | 34.09% | 18.46% |
| Education | 0.00% | 10.77% |
| Language barrier | 9.09% | 6.15% |
| Diagnostic testing | 0.00% | 1.54% |
| **Global** | | |
| Brand | 6.82% | 7.69% |
| Supply chain | 2.27% | 3.08% |
| **Other** | | |
| Other | 0.00% | 1.54% |
| **None** | | |
| None | 9.09% | 6.15% |

particular medications "*depend on season. This season [December] getting to harmattan, [so there is] a lot of cold medication prescription, with headache, cold, nausea, runny nose. In rainy season there is a lot of malaria because mosquitoes multiply.*" The most prevalent challenge mentioned at the level of the shops was in fact low sales volume. Specifically, 13.64% of challenges fell under this category in herbal and chemical shops among all challenges, while comparatively, pharmacies did not mention this concern. We note that unlike wage-based employees, a third of respondents in herbal and chemical shops are shop owners who presumably directly benefit from the proceeds generated by the shop.

The global COVID-19 pandemic was also implicated in the reduced customer traffic and sales. Fearing both a potentially infectious environment and the social and economic repercussions of a positive COVID diagnosis, many would-be customers avoided sites they associated with sickness. Shops' proximity and patients' differing access to hospitals and clinics were reported to influence customer traffic and behaviors. A respondent in a pharmacy replied "*Sales are low. Malaria symptoms are just like COVID symptoms. If [they] have malaria, just similar to COVID—cough, sneeze, feverish, just the symptoms of COVID. It's difficult because of this COVID. It interrupted a lot of pharmacies because [would-be customers are] afraid to come for drugs*" (35–44 year old female chemical shop owner in a medium-income neighborhood). The shelf life of medicines was related to low sales as a function of the supply chain. Specifically, medicines arrived at or close to their expiry date, and with low sales traffic would expire before they could be sold, e.g., as one said: "*expiry of drugs; when drugs arrive they are almost expired and don't move fast*" (35–44 year old male pharmacist in a low-income neighborhood).

Competition, especially among pharmacies, was also relevant to the discussion of low sales. Many shops were observed to cluster around hospitals, and customers often looked for the

best price, which may have been available at nearby shops. Customers' purchasing decisions also appeared to be made based on price above any other factor. Customers typically started at the pharmacy closest to the hospital, inquired about prices, and moved down the line to adjacent medicine sellers until they found the lowest price.

**Workplace.**   Among the workplace-related challenges cited by respondents were insufficient income, and boredom. These fell under the broader umbrellas of job satisfaction and issues of limited resources mentioned in the existing literature. For instance, one respondent expressed "*it gets so boring doing the same thing over and over again*" (25–34 year old male pharmacist in a high-income neighborhood). Based on prior work, experiences within a shop are expected to influence the quality of care provided [43], however, as noted above a relative minority expressed such concerns as the most important challenges they face. Individual challenges, such as insufficient income, may, therefore, play into enthusiasm for a job and the capacity of staff to provide high-quality care. The effect may transcend customers, as pay may also affect the likelihood of attrition, a concern highlighted earlier.

The lack of feedback mechanisms, especially as they relate to the provision of factual information, also presents a challenge to shopkeepers. Keeping informed is an undertaking beyond the conventional job description. Without systematic patient education and monitoring, effective antimicrobial stewardship relies on practices beyond the typical duties of respondents, which many have limited experience with. In such contexts, the scope of service of medicine sellers has been observed to depend on individual motivation and perceptions of legitimacy [16]. Thus, the job satisfaction and motivation of these distributors, and their ability to make ends meet from the pay they receive at the individual level, may be considered particularly salient.

## Challenges at the customer level

Respondents in our study occupy patient-facing roles, and their conversations with customers influence customers' subsequent health-seeking behaviors. Specifically, patients' choice of quality medicines and adherence to treatment often depend on the information that they receive at the time of purchase. Shopkeepers' perceptions of illness, in turn, are also informed through interactions with customers [48]. Health outcomes are also shaped by patients' decision-making and their choices of care [49]. We organize the challenges in the customer category into the four subcategories, which we discuss in turn.

**Customers' attitudes.**   Customer attitudes were the most prevalent challenge mentioned by pharmacies, and were also prominent in herbal and chemical shops. The role of customers was often described in specific terms, or indirectly, by focusing on the need for a great deal of patience. A difficulty in addressing this challenge is the notion of "the customer is always right" and how this idea influences responses to customer attitudes. Customers' attitudes, however, should not be considered in isolation. The heightened emotions associated with illness were reported to exacerbate challenges, as articulated by one: "*People walk in with anger. When sick, people become frustrated. They ask for a certain drug. . . Some may insult or say mean things. . . I try to advise, but they don't want to be told what he/she should consume*" (35–44 year old female MCA in a pharmacy in a middle-income neighborhood). Although the attribution of attitudes occurs in this fashion in our data, it is important to note that this challenge is only mentioned 35% of the time in isolation. In the remaining 65% of cases when it is mentioned, respondents reference challenges with payment in an overwhelming majority of cases, where customers are short on money, can't afford the medications prescribed or recommended to them, or attempt to buy on credit that is not extended to them. For example, as one respondent connected these issues, they said: "*insults. . . They normally complain of prices,*

*aside that, nothing else [I find challenging]*" (18–24 year old female medical counter assistant in a medium-income neighborhood). Multiple respondents also reported customers' resistance to their inquiries into symptoms or dismissing their advice and linked this to customers' attitudes. Regarding their stewardship of antibiotics, one pharmacist reported "*Some patients don't understand antibiotics for instance. [Customers say] 'I am buying, [I] don't understand why you don't want to sell it to me.' Sometimes it turns into an argument*" (25–34 year old male pharmacist in a low-income neighborhood).

**Self-diagnosis and self-medication.** The practice of self-diagnosis was reported to be widespread. Customers subverted the process of going to hospitals or clinics and instead identified an illness themselves based on the symptoms they perceived relevant. Efforts to self-medicate with over-the-counter medications often brought customers to medicine sellers. This was framed as a challenge when respondents disagreed with patients' self-diagnoses. Self-medication was mentioned explicitly as a challenge by 10 respondents (12.31% of reasons in pharmacies), notably concentrated in pharmacies, rather than chemical shops or herbal shops. When customers did not have prescriptions, but believed they needed medications such as antibiotics, respondents reported difficulties: "*customers tell 'I want this' but this medicine will not solve their problem. Sometimes they will not understand why, but I say no [to their request] if I know it won't help them*" (35–44 year old female MCA in a pharmacy in a middle-income neighborhood). Respondents sometimes alluded to their conditional allowance of customer self-medication, perhaps offering medications without prescription if they deemed the chosen treatment course to be useful to the customer.

**Adherence.** Completing a full course of antimicrobials is essential for effective treatment and preventing resistance. Respondents reported general uncertainty about customers' completion of treatment courses and a lack of clarity on the state of antimicrobial resistance in Ghana. To overcome this lack of centralized guidance, one pharmacist reported independently reading scientific articles to remain informed about current best practices, and others said they were doing research on the Internet. Respondents explained that AMR was not one of their main considerations before distributing drugs, but some seemed to be concerned about the rapid emergence of drug resistance that renders their products obsolete.

In a separate question, 41% of respondents reported that they believed that patients "probably" or "definitely" adhered to their entire antimicrobial treatment courses, with one attributing this trust to the fact that "*it is expensive, no one wants to waste money*" (25–34 year old female MCA in a pharmacy in a medium-income neighborhood). 54.8% answered "might or might not," and the remaining respondents expected that customers would not complete their treatment courses. Adherence monitoring was described as being carried out at the individual level. When asked "Do you think customers actually finish the entire treatment course of antibiotics after they leave the shop?" one responded "*Definitely yes, because I talk to them. I tell them when you don't finish up with the course, your treatment will not wash off [the infection]*" (35–44 year old female chemical shop owner in a medium-income neighborhood). Another reported that they took phone numbers and routinely called patients to verbally ensure that they finished their antibiotic course. The majority had doubts, echoing the respondent who said "*Not everybody takes everything. Even though you tell them to finish the course, you realize sometimes somebody does not take the whole course, has 2 days left*" (25–34 year old male pharmacist in a high-income neighborhood).

The issue of adherence is closely related to self-medication, and was mentioned by respondents both in herbal and chemical shops, as well as in pharmacies. This issue was highlighted in herbal or chemical shops relatively more frequently (6.82%) compared to pharmacies (1.54%). Some respondents mentioned the potential for drug abuse; one believed that it occurs "*especially [for] antibiotic and analgesic*" medications (25–34 year old male pharmacist in a

middle-income neighborhood), and another explained that they responded to such situations by dodging the customer's request: "*If I think they will abuse it, I tell them it is out of stock*" (35–44 year old female MCA in a pharmacy in a middle-income neighborhood). Although it was unclear what was explicitly interpreted as "abuse," the context of antimicrobials suggests that abuse could involve either taking an excessive dosage at once, not taking all prescribed medication, or using medicines for effects other than their prescribed purpose.

The inconsistent availability of dosage strengths and treatment options was described as a challenge for some respondents. Medications that required frequent dosages (such as every four hours, or tablets with small dosages that required taking up to four pills at each interval) were reportedly disliked by customers. Some customers would decline to purchase these medications, and respondents expressed concern that patients who did purchase these typically lower-cost medicines likely would not adhere to their regimens.

**Role incongruence in treating customers.** The role of medicine dispensary employees in the diagnosis and treatment of disease was also considered. Treatment-related issues came up as challenges in a variety of ways, and were mentioned more prominently in herbal or chemical shops compared with pharmacies. These mentions included role-conflict, where customers perceived respondents as physicians. Some respondents noted that they lacked the necessary training to respond to their customers' needs. This was especially salient when facing complicated symptoms or navigating situations without diagnostic testing to establish the appropriate course of treatment. These instances and an absence of information often impeded respondents' ability to tailor their advice and dispense medicines. A respondent reported that "*People consider you as a doctor, see you as a doctor. It becomes very challenging if you make a mistake because you'd be held responsible*" (45–54 year old female pharmacist in a high-income area).

The disparity between licensed roles and actual practice thus came up as a challenge. Respondents suggested that formal medical institutions were not able to meet the health needs of the general public. As a result, they found themselves asked to fill roles beyond simple medicine distribution. For instance, customers without prescriptions sometimes sought diagnosis within medicine dispensaries. Although some could be easily treated, such as malaria with rapid diagnostic testing, others were less straightforward. Facing complicated symptoms was mentioned as a challenge in this context: "*facing difficult cases of illnesses that I cannot help with,*" (45–54 year old female chemical shop owner in a low-income neighborhood) that underscores gaps between distributors' individual knowledge and the demands of a fragmented healthcare ecosystem.

## Challenges at the community level

The broader circumstances of the communities surrounding medicine vendors are likely to affect the experiences and challenges respondents faced [47]. In the socioecological model of public health, the community category could include factors descriptive of the surrounding community and identified as barriers to optimal stewardship. Among all challenges mentioned, the Community category represented 44.95%. Specifically, community-level barriers accounted for 50.00% of challenges listed in chemical and herbal shops, as well as 41.53% of pharmacies identified these barriers. We discuss below four subcategories that emerged from our analysis.

**Payment.** As described by staff at dispensaries, paying for medications was more than a mild concern for customers. The cost of medications was an obstacle for the majority of their clients. Although cost was always a large barrier for customers, prices varied widely by shop. Individual medications were priced according to the brand and origin of the medication production. The cost of antimalarial medications was typically lower for generic and locally

produced, relative to imported, brands. In general, prices for medicines varied from shop to shop. One respondent mentioned that a "*patient might return with the same condition just because they couldn't afford a drug*" (25–34 year old male pharmacist in a low-income neighborhood) and another reported "*after you finish giving out the medicine, he/she will be like 'the money is too much, I can't afford, your things are too expensive,' and the person will just go out without buying. This mostly happens, happens often*" (45–54 year old female MCA in a medium-income neighborhood).

Financial stress was reported to affect the relationships cultivated between respondents and their customers, and manifested in different ways, such as attempting to buy on credit. Other respondents reported yet different strategies, such as "*customers trying to return sold drugs*" (55–64 year old female chemical shop owner in a middle-income neighborhood). Most shops explicitly specified that they required payment immediately at the time of sale. Some individual shopkeepers occasionally trusted customers with credit allowances to pay later, which was more common for repeat customers. One reported that for patients who were "*struggling and can't pay, I give and pay the rest out of my own funding*" (18–24 year old female MCA in a chemical shop in a medium-income neighborhood).

**Education.**   Many shopkeepers sought to influence the health-seeking behaviors of their customers, often by advising and informing them of medicines that would be most useful given their condition. In some cases, these workers viewed their role as countering the (mis) information circulating among the general public. In other cases, they mentioned that the illiteracy or lack of education of the typical customer created a challenge when interacting about the proper course of treatment. They noted, in their view, that illiteracy and a lack of knowledge at the community level were almost insurmountable to fix. This lack of education may have been a driver of other challenges, such as self-diagnosis. One respondent explained that "*people coming in with self-diagnoses and the medications they want, sometimes without considering your recommendations or opinions*" created their biggest challenge, thus "*trying to get them to understand why they don't need an antibiotic for a very trivial issue*" (18–24 year old pharmacist in a middle-income neighborhood). Education was pointed out by pharmacists in particular as one in ten challenges faced. Coincidentally, this challenge was not mentioned by respondents in chemical shops or herbal shops. Given the differences in background and training of medicine distributors by shop type, it is possible that pharmacists and trained MCAs perceived this challenge more acutely, and suggests that this challenge may also be present in herbal or chemical shops, albeit not at the forefront of concerns for shopkeepers there.

**Language barriers.**   Some respondents reported a language barrier, specifically when working in neighborhoods with large communities known for being hubs of in-migration. This was framed as a challenge for understanding and responding to medical conditions. Specifically, language barriers represented 9.09% of all challenges in herbal or chemical shops, and 6.15% of all challenges reported in pharmacies. These challenges may have been more manageable for long-serving respondents (who were far and few) who had established rapport in their communities. This underscores again the importance of community interactions in shaping the gravity of challenges perceived by respondents, and should be considered in light of the relatively high turnover reported earlier.

**Diagnostic access.**   Some stores reported that they did not offer diagnostic testing for malaria because tests could be accessed at a hospital nearby. However, such an assumption may mean that customers who cannot afford hospitals do not have access to on-demand diagnostic tests in lower-cost medicine shops. Shops located closer to hospitals faced less difficulty with customers, as their patients were more likely to have undergone diagnostic testing and bring prescriptions. Conversely, shops farther from hospitals struggled with a lack of access to diagnostic resources. One respondent reported as their primary challenge the insufficient

"*availability and accessibility of diagnostic tests and diagnostic facilities for customers or patients*" (25–34 year old pharmacist in a middle-income neighborhood). The lack of diagnostic testing may have manifested in other challenges, such as self-diagnosis and self-medication, discussed in the Customer category.

## Challenges at the global level

Many challenges in the present context have international origins. When articulating their main challenges, respondents cited supply chain issues, and made references to conflicts around the brands of medications. There were no variations by shop type, as the relative frequencies they appeared are comparable across herbal or chemical shops and pharmacies.

**Brands.** The role of "brand" was considered important in decisions around the efficacy of medicines. Some respondents voiced a stronger belief in the efficacy of imported or originator brands than local generic brands, with one saying "*you can never compare generics to innovator, [imported originator brands being] more efficacious*" (25–34 year old male MCA in pharmacy in a middle-income neighborhood). Many customers shared these views, and reportedly established preferences for particular medications based on prior experiences, advertisements, or reports from acquaintances about which medications effectively cured their ailments. Two respondents specifically reported as their biggest challenges the television and billboard advertising that they believed shaped customers' preferences in this way. One explained that their biggest challenge was "*trying to convince a patient to buy another type of drug aside from what he or she thinks is good or what a layperson or TV/radio advert [advertisement] has explained*" (18–24 year old male healthcare assistant in a chemical shop in a high-income neighborhood). These respondents perceived a responsibility to work against advertisements (and other laypeople), believing that their expertise was better informed than market forces such as advertising.

**Supply chains.** Part of the reason for lacking certain medications is related to supply chain issues. Herbal or chemical shops sold a variety of artemisinin-based combination therapy (ACT) antimalarials, and pharmacies tended to sell a variety of both imported and local brands of antibiotics. Respondents described varying degrees of confidence in each brand, but could not always procure the preferred variety, as articulated by the respondent who said "*sometimes the medicine they want, we don't have it*" (25–34 year old male employee of a pharmacy in a high-income neighborhood). As an example, when one respondent explained their biggest challenge they said "*In my shop, for instance, I think probably the lead time of the medications. And actually, sometimes the drugs you request that you want, they don't really come. They [the wholesaler] will just give you what they have in their warehouse*" and cited the challenge of "*owing suppliers due to low sales and expiry of drugs*" as a corollary (25–34 year old female pharmacist in a medium-income neighborhood). We note that some shops did not experience delay between running out of a medication and resupplying, and a further 30 reported that medicines could be restocked within a matter of days.

**Challenges of a global nature expected but not mentioned.** Prior studies suggested that the lack of public health policy or lack of enforcement posed a challenge. Earlier studies have reported that in the present context, policy enforcement is sporadic and health regulation infringement is frequent [31, 50]. Indeed, respondents reported little faith in regulations to improve their practices and identified challenges with potential policy implications, i.e., low diagnostic capacity, increasing prices, difficulties with supply management, and uncertain drug quality assurance. No respondent referenced national or municipal regulation as a strategy to alleviate their challenges, as laws or inspections would more likely be seen as impediments to business. In contrast, respondents expressed general disillusionment with policy-

making and the absence of expectation of government support or action on the concerns they face.

## Discussion

Employees in pharmacies and chemical shops create an informal system of healthcare at the neighborhood level. They perform a vital role in communities as providers of medical care and fill in gaps left by larger institutions in Accra, Ghana. In their role, however, they find several obstacles that truncate their ability to fully meet the needs of patients. These challenges, unfortunately, also at times prevent medicine sellers from being responsible stewards of antimicrobial medications, and might thereby contribute to the incidence of AMR.

Echoing findings of prior studies [51], medicine vendors indicated that their customers' attitudes posed significant challenges in service provision. Our analysis suggested that this was tied to an equally prevalent obstacle faced by medicine vendors: the ability of their patients to afford medicines. That is, frustrations from not being able to afford medicine may influence the type and quality of customer-medicine vendor interactions. Cost was also reported to supersede other factors and shape how customers interacted with staff at dispensaries. A customer who sought to purchase partial treatment due to financial constraints may exit a shop without making a purchase, if their request for subtherapeutic treatment is refused. Medicine vendors' efforts to practice antimicrobial stewardship may in turn have financial implications for shops.

Both interpersonal and cost-related challenges were evident at all levels of our modified socioecological model of public health, and manifested through a variety of linked issues. Importantly, originator brand medications (as opposed to generics), have been highlighted as a point of conflict. On the one hand, customers demanded brand-name medicines, but they were not necessarily able to pay for them, relative to generics with the same active ingredients. On the other hand, medicine sellers do not have the infrastructure in place to confirm the quality of medicines [52–54], which may have been partially driving beliefs (of both customers and medicine sellers) that generics were less effective than international brands. This made medicine sellers less effective when advocating for generics on the basis of active ingredients than they could have been otherwise. Moreover, not all shops carried originator brand medications either due to supply chain issues or perhaps out of fear of low sales that were frequently mentioned as a challenge. This was especially the case in herbal and chemical shops. As articulated by medicine vendors, these interlocking challenges caused difficulties in effectively counseling patients either because of customers' hesitation to consider their recommendations or because of other barriers, such as customers' knowledge of differences or similarities between generic and originator brands.

Policy changes in Ghana in response to AMR have focused on restricting the availability of antibiotics through prescription laws, which are often not strictly enforced. Existing literature noted that vendors distributed antimicrobials regardless of any laws in place, and asserted that doing so was a widely known expectation among the general public [28, 55]. Our findings, however, suggest that the most significant challenges driving AMR are not fully addressed by the antibiotic prescription laws. While stricter enforcement may ameliorate the problem, it is unlikely to be a panacea, given the multiple levels of barriers that drive AMR. Counter to some existing literature, we found that medicine vendors were judicious about antibiotics and antimalarials, and expressed an overall desire to safeguard medications [25, 28, 55, 56].

The broader trend of economic downturn in Ghana due to COVID-19 [57] intersected with medicine dispensation in ways that were not expected, highlighting the vulnerabilities in such contexts. Because of COVID restrictions, the general public could not access healthcare

effectively and avoided overcrowded hospitals during the outbreak. As such, an increased reliance on pharmacies and non-institutional settings for seeking healthcare could have solidified further [58]. COVID-19 behavior changes may have shifted dispensaries towards being the first point of contact for individuals seeking medical care. These challenges, including those identified in this study, are not static, and the vulnerabilities brought to light by the COVID-19 pandemic may have exacerbated these challenges.

Overall, while these challenges do not come with "quick fixes," working with shopkeepers and placing value on their knowledge and experiences, to further promote medicine stewardship would be an opportune next step. Stewardship-advancing work processes found in other contexts [17] can be implemented in Accra. For example, previous studies showed that targeted investment in the systematic education of community health providers reduced knowledge gaps [59, 60]. Other strategies, including making diagnostic testing available at no cost to patients or medicine vendors, and reducing financial inaccessibility of high-quality full-course medicines, could help improve service delivery and patient outcomes.

## Limitations

Social desirability bias and the potential legal ramifications may have led participants to give idealized responses to appear to be compliant with expected standards. We did not ask chemical shops about antibiotics because they are not legally sanctioned to sell them. Thus, we did not capture illicit antibiotic sales that may significantly shape the community health landscape. However, the challenges described by the shopkeepers are believed to be generally consistent across different types of medicines.

As reported by participants, disease outbreaks vary seasonally. However, this study was limited to November–December, in between the rainy and harmattan seasons. It thus provides a limited lens into seasonal variation, and reports about antimalarials are based on recall. Additionally, respondents' beliefs about consumers' behaviors after purchasing medications and leaving their shop may be inaccurate. In addition, as in all contexts, survey questions may have been interpreted differently by different respondents. The questionnaire was designed to build upon and corroborate answers to interrelated questions, but the potential for misinterpretation and mistranscription remains.

To minimize the risk to subjects, we did not explicitly ask about illegal antibiotic distribution. Thus, inferences were made and extrapolated from available responses, such as questions focused on over-the-counter antimalarials. Although this is a clear limitation, it is crucial to note that respondents in herbal shops or chemical shops did not bring up self-medication or self-diagnosis as a major challenge that they navigated. This gives us confidence that this limitation is unlikely to mask important issues that we were unable to uncover.

It is important to note that the underlying causes of challenges may reside in different tiers of the socioecological model of public health than the one that respondents or researchers identified. Issues identified at the Entity, Customer, or Community levels may in fact be indicative of deeper systemic barriers, and should be addressed as such. In contrast, challenges may have been perceived as entity-specific even if root causes could be extrapolated more broadly.

To verify the findings of this study, future work should elucidate both the supply- and demand-side drivers of antimicrobial use. Further validation of these findings should be undertaken in future studies, by conducting the survey in additional shops, as well as complementing this with focus groups with medicine vendors to aid interpretation. Exit interviews with customers should verify shopkeepers' assessments, and the use of and "mystery patients" could systematically probe medicine-selling behaviors. Furthermore, future studies could seek to verify data with multiple employees of each shop.

## Recommendations

While keeping in mind the limitations of this study, we believe that the insights from shop-keepers provide valuable information to improve the quality of care in private medicine dispensaries. The systemic factors leading to the discussed challenges—poverty, lack of education/knowledge, and faltering infrastructure—extend far beyond any quick fixes. However, ameliorating some of the conditions surrounding medicine shops would contribute to more prudent antimicrobial stewardship. These challenges could elucidate some possible interventions to enable quality-assured diagnoses, well-stocked pharmacies, and patient-focused health care. Future studies should aim to prioritize our recommendations, supported by cost-benefit and cost-effectiveness analyses. It is beyond the scope of our project to prioritize the recommendations we mention below or to compare them in terms of associated cost. We however provide these recommendations, as they point beyond the long-lasting systemic issues that impact not only healthcare, but other facets of everyday life in Accra.

1. **Centralized training for staff in chemical shops and pharmacies:** Comprehensive training programs that couple information about antimicrobial resistance with strategies to navigate typical daily challenges could prove invaluable to increase the motivation of these staff to practice stewardship. This training could help MCAs feel more secure in their role, alleviate the burden of individually seeking this information, and perhaps address some of the day-to-day frustrations they face on the job.

2. **Free rapid diagnostic testing:** Providing confirmatory diagnostic testing without additional fees appears extremely important for ensuring the judicious use of antimicrobials. Increasing the availability of tests and subsidizing fees for their users could enable a more informed treatment of malaria based on diagnoses rather than symptom-based presumptive treatment. This may require more laboratories or subsidies for rapid in-store diagnostic tests. Note, however, that a replacement effect may occur, in which distributors default to antibiotics for cases with negative malaria test results as in [61].

3. **Quality assurance for customer trust:** Respondents reported that many customers prefer to use only brands with which they are familiar, based on previous experience, advertising, or recommendations from trusted acquaintances. The general public may recall stories of counterfeit and substandard medicines in West African markets [26, 62], driving valid concerns expressed by many when offered locally sourced medications. Therefore, regular checks of all medications and publicity for such checks could enhance trust in medications that are high quality, cheaper, and locally produced. This would include enforcing mandatory minimum strengths for medicines that could help prevent the use of subtherapeutic doses by making sure all medicines in the inventory are sufficiently efficacious.

4. **Include the active ingredients in advertising:** Encouraging the listing of active ingredients in advertisements could help customers recognize the legitimacy of medicines offered through their active ingredients. Making customers familiar with active ingredients directly, coupled with more tightly regulated monitoring, may assuage customers' fears and could help promote local manufacturing.

5. **Enhancing/assuring shelf stability:** The development of more shelf-stable medications can help mitigate supply-side issues, as could a consignment system for manufacturers to replace drugs that have expired. Given that none of the stores visited appeared to have refrigerators for medications and few were air-conditioned, it is possible that their expiration may occur faster than expected. Thus, it should be ensured that medicines are shelf-stable without refrigeration.

6. **More robust reporting pathways:** Based on respondents' reports, it is apparent that centralized monitoring and a unified information source for antimicrobial resistance should be developed for physicians, pharmacists, and MCAs to communicate when antibiotics do not work or indicate projected drug resistance. Streamlined routes for reporting side effects could also improve customer trust and promote the availability of safe and effective treatments.

## Conclusion

Communities without access to formal healthcare appear to seek low-cost pharmaceuticals rather than consulting physicians in larger institutions. Dispensaries in turn shoulder the burden, and in the process encounter many challenges, which span multiple levels. The Customer- and Community-level tiers of a socioecological model of public health presented the most profound challenges for medicine dispensaries in Accra. Notwithstanding the dynamic set of challenges, antimicrobial misuse appears to be a result of systemic and recurring gaps in the ability to access effective medical treatment. Future studies must continue to examine the importance of medicine distributors in the community [63], but should also undertake an approach that takes into consideration the customer, global, and related context.

## Supporting information

**S1 File.**
(PDF)

## Acknowledgments

We thank the participants of this research who articulated their experiences and generously shared time out of their workdays, going above and beyond to explain the landscape of public health through their eyes.

## Author Contributions

**Conceptualization:** Hannah Camille Greene, Kinga Makovi, Jemima A. Frimpong.

**Data curation:** Hannah Camille Greene, Kinga Makovi.

**Formal analysis:** Hannah Camille Greene, Kinga Makovi, Akhil Bansal.

**Funding acquisition:** Hannah Camille Greene, Akhil Bansal.

**Investigation:** Hannah Camille Greene, Rafiatu Abdul-Mumin.

**Methodology:** Hannah Camille Greene, Kinga Makovi, Rafiatu Abdul-Mumin, Jemima A. Frimpong.

**Project administration:** Hannah Camille Greene, Kinga Makovi, Rafiatu Abdul-Mumin, Jemima A. Frimpong.

**Supervision:** Kinga Makovi, Akhil Bansal, Jemima A. Frimpong.

**Validation:** Hannah Camille Greene, Kinga Makovi, Rafiatu Abdul-Mumin, Akhil Bansal, Jemima A. Frimpong.

**Visualization:** Hannah Camille Greene.

**Writing – original draft:** Hannah Camille Greene, Akhil Bansal.

**Writing – review & editing:** Hannah Camille Greene, Kinga Makovi, Akhil Bansal, Jemima A. Frimpong.

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
