## [Decision Letter · Decision Letter 0]

24 Apr 2023

PONE-D-23-02556Challenges in the distribution of antimicrobial medications in community dispensaries in Accra, GhanaPLOS ONE

Dear Dr. Frimpong,

Thank you for submitting your manuscript to PLOS ONE. After careful consideration, we feel that it has merit but does not fully meet PLOS ONE’s publication criteria as it currently stands. Therefore, we invite you to submit a revised version of the manuscript that addresses the points raised during the review process.

We look forward to receiving your revised manuscript.

Kind regards,

Francesca Baratta, PharmD, PhD

Academic Editor

PLOS ONE

“Above all, we thank the participants of this research who shared their experiences and

generously shared time out of their workdays, going above and beyond to explain the

landscape of public health through their eyes.

This fieldwork would not have been possible without Rafia Abdul-Mumin, who

conducted 40 fieldsite visits. Her insight and hard work were invaluable, and the success

of the study is truly indebted to her.

The fieldwork for this project was funded by the Social Sciences Division at NYU

Abu Dhabi, and the publication was funded by a grant from the Effective Altruism

Infrastructure Fund.”

“The research fieldwork was funded by the New York University Abu Dhabi Division of Social Science as an undergraduate thesis project (of HG) (https://nyuad.nyu.edu/en/academics/undergraduate/undergraduate-research/capstone.html). Subsequent data analysis and writing were funded by a grant received by HG (https://av20jp3z.paperform.co/?fund=EA Infrastructure Fund) from the Effective Altruism Infrastructure Fund (https://funds.effectivealtruism.org/funds/ea-community).  The funders had no role in study design, data collection and analysis, decision to publish, or preparation of the manuscript.”

6. Please remove your figures from within your manuscript file, leaving only the individual TIFF/EPS image files, uploaded separately. These will be automatically included in the reviewers’ PDF.

Reviewers' comments:

Reviewer's Responses to Questions

**Comments to the Author**

1. Is the manuscript technically sound, and do the data support the conclusions?

Reviewer #1: Yes

Reviewer #2: Yes

2. Has the statistical analysis been performed appropriately and rigorously? 

Reviewer #1: N/A

Reviewer #2: Yes

3. Have the authors made all data underlying the findings in their manuscript fully available?

Reviewer #1: Yes

Reviewer #2: Yes

4. Is the manuscript presented in an intelligible fashion and written in standard English?

Reviewer #1: Yes

Reviewer #2: Yes

5. Review Comments to the Author

Reviewer #1: The paper assesses context-specific attitudes and challenges faced by community dispensers in Accra, Ghana, in their sale of antimicrobials. This is relevant in promoting the rationale use of antimicrobials in order to limit the development and spread of drug resistance.

Some comments to improve on paper quality.

Introduction:

1. Lines 25 - 26: Authors should consider providing current data on NTDs and malaria. The references are old and may not give a true picture of current trends. It is hard to believe that Accra, being the capital of Ghana, will have many NTDs. The paper referred to actually talks about non-communicable diseases in Accra and not NTDs.

Methods:

1. Lines 65 - 67: Consider revising to "Medicine vendors are reported to be the primary source of medicine dispensation in Accra, Ghana, and the whole of West Africa" . The current sentence gives the impression of ranking with Accra leading

2. Sampling:

- Authors should provide details of how the sampling of shops was done (sampling frame etc)

- What were the regions of Accra considered (line 91)?

- Any details on distance from hospitals considered?

- Most pharmacies will have multiple employees. How were the interviewees selected? Was one interviewee selected in each shop?

3. Provide detailed descriptions of the different types of staff in the shops.

Results:

1. Line 176: Provide references for the previous studies

2. Lines 179 - 182: This is not clear. Where are these under the individual circle in figure 1?

3. Lines 183 - : Quantify "most", "many" etc with proportions

Table 1:

1. What is the professional background of the shop owners? If authors are interested in ownership status then that should be considered separately. Ownership cannot be put under the same variable as professionals.

2. What is the basis for the staff categorization? What is the difference between the health care assistant and the MCA? Who is a herbal employee?

3. Authors should clarify what missing gender means. It's not clear how gender was missed when participants were visited.

4. Authors should define the different categories under source of employment. It's not clear how source of employment will be unknown when handling a sample of < 100.

Table 2:

1. "Year of starting to work in this location" should be separated from the rest of variables in the table. Mean year with SD etc appears misplaced. Authors should consider proportions of shopkeepers starting work in each of the given years (2014 - 2021) per type of shop keeper.

Generally, can authors relate results to type of shopkeeper interviewed in terms of role, age-group, gender, years of work etc. Will be interesting to know what the different types of shopkeepers are saying. This will inform any planned interventions.

Reviewer #2: I would like to thank the authors for conducting this vital study. Please find my comments below:

Abstract:

I recommend the removal of both hyphens in the sentence in the abstract which says: ‘Yet patterns in medicine distribution—and their consequences—are not well understood’.

Secondly, I recommend elaborating on the methodology in the abstract section. The current information is skeletal and does not to provide sufficient information on methods applied. As I understand you may be constrained by the word count, I recommend shortening the introduction to achieve this.

Thirdly, the authors state a two-fold objective. However, the results in the abstract appear to focus on only the first objective (challenges). I am wondering if authors identified opportunities for improving community-level stewardship of antimicrobials from the survey and I recommend including these in the abstract.

Main Article:

I recommend the removal of hyphens in lines 10 and 11, and between words in other parts of the manuscript.

Most importantly, I recommend a thorough review of the introduction to ensure there is a flow of ideas culminating in a comprehensive study rationale. The current introduction appears to consist of distinct parts with two study rationales in lines 20-23 and 58-59 making the research ideas appear quite disjointed. The introduction should be more concise and integrate both ideas. This should be followed by a comprehensive research rationale at the end.

Lastly, the study was conducted in Ghana but approved by the New York University, Abu Dhabi Institutional Review Board. There is no mention of ethical approval with any Ghanaian institution. I am wondering if you such approval was obtained.

6. PLOS authors have the option to publish the peer review history of their article (what does this mean?). If published, this will include your full peer review and any attached files.

Reviewer #1: No

Reviewer #2: No

---

## [Author Response · Author response to Decision Letter 0]

3 Aug 2023

We sincerely thank you for your guidance and requested edits. The Response to Reviewers is uploaded as an attachment as requested.

---

## [Decision Letter · Decision Letter 1]

14 Sep 2023

PONE-D-23-02556R1Challenges in the distribution of antimicrobial medications in community dispensaries in Accra, GhanaPLOS ONE

Dear Dr. Frimpong,

Thank you for submitting your manuscript to PLOS ONE. After careful consideration, we feel that it has merit but does not fully meet PLOS ONE’s publication criteria as it currently stands. Therefore, we invite you to submit a revised version of the manuscript that addresses the points raised during the review process.

We look forward to receiving your revised manuscript.

Kind regards,

Francesca Baratta, PharmD, PhD

Academic Editor

PLOS ONE

Reviewers' comments:

Reviewer's Responses to Questions

**Comments to the Author**

1. If the authors have adequately addressed your comments raised in a previous round of review and you feel that this manuscript is now acceptable for publication, you may indicate that here to bypass the “Comments to the Author” section, enter your conflict of interest statement in the “Confidential to Editor” section, and submit your "Accept" recommendation.

Reviewer #1: All comments have been addressed

Reviewer #3: (No Response)

Reviewer #4: (No Response)

2. Is the manuscript technically sound, and do the data support the conclusions?

Reviewer #1: (No Response)

Reviewer #3: Partly

Reviewer #4: Partly

3. Has the statistical analysis been performed appropriately and rigorously? 

Reviewer #1: (No Response)

Reviewer #3: N/A

Reviewer #4: No

4. Have the authors made all data underlying the findings in their manuscript fully available?

Reviewer #1: (No Response)

Reviewer #3: Yes

Reviewer #4: Yes

5. Is the manuscript presented in an intelligible fashion and written in standard English?

Reviewer #1: (No Response)

Reviewer #3: Yes

Reviewer #4: No

6. Review Comments to the Author

Reviewer #1: (No Response)

Reviewer #3: Thank you for inviting me to read this revised submission based on previous reviews.

To curb the global threat of AMR, strengthening community antibiotic stewardship could contribute much to rational use of antibiotics, especially in settings where antibiotics are also available without a quality assured medical diagnosis and prescription. As such, it seems a relevant topic to assess challenges of community medicine providers in Accra.

I still have some concerns however.

The primary objective was to understand challenges perceived by medicine vendors. It is assumed that if these could be addressed, it would contribute to health of the population. However, this is not a given. First, the challenges identified are beyond quick fixes, related to poverty, knowledge, infrastructure. Second, to improve the health outcome of the population attending these vendors, it is key that a quality assured diagnoses is made (requiring proper training and refresher courses for staff; quality assured diagnostics), that respectful quality care is delivered (again requiring proper training and refresher course, as well as validated guidelines with empiric antibiotic choices based on the best available surveillance data) and that antibiotics which are being sold are quality assured and well stocked. Several of these issues are mentioned in the discussion already (but without a clear pathway or prioritisation to bring about such major investments). Thus, while addressing reported challenges might increase job satisfaction of the staff, and while there is great potential in involving this group as gatekeepers to improve health and antibiotic use, it remains to be tested to what extent this results in better care and in better use of antibiotics. The authors need to acknowledge this limitation throughout.

The paper should provide more details on the population being sampled, also to provide insight in potential selection bias. In particular, it is stated that all community providers were invited, but that saturation was not achieved, with only 80 included. It is essential to provide details on the total number of each provider (pharmacy, MTC, herbal shop, but potentially also other medicine providers, such as churches, traditional healers, (mobile) market sellers, ..) in the areas, and how a selection was made. For the 17 neighbourhoods (selected by convenience): please clarify what is their population, what % of the population of Accra is covered. In the discussion, reflection on how these non-random selections may have biased results needs to be included. Also, please clarify the turn-over of staff: Table 1 states 42.5% less than 2 years in the position; the discussion mentions over 50%.

Furthermore, methodologically it should be clarified why only individual open-ended questionnaires were used to extract perceived challenges, rather than also bring people together with some (curated) focus group discussion to attain validation and more in depth knowledge.

It would have been very relevant to have data on actual health seeking behaviour of the populations in the 17 catchment areas (who is attending which providers when; how much health care is actually provided by medicine vendors), but this was presumably beyond the scope of this study. Also, an extension with exit interviews of customers, or the use of mystery patients, could have provided validation of the self-reported dispensing of the vendors. Please consider if this should be included in the recommendations.

Another potential source of bias is related to the timing of the study: only 2 months were selected. Demand for/use of antibiotics and antimalarials tends to be highly seasonal, and also health care seeking behaviour and accessibility can be impacted by seasonality. Please acknowledge and reflect on this limitation as well.

The 120 reported challenges in the 80 interviews have been grouped in 5 categories. The specific challenges are presented grouped in Figure 1, but otherwise reported in 5 rather long-winding narratives. This is not very accessible, therefore I strongly suggest that each of the narratives is clearly structured with subheadings along the 2-5 subcategories as presented in Figure 1.

Antibiotic use is the result of a demand from customers and supply by providers. As mentioned, it is unfortunate no information is available on the population which was seeking care/antibiotic use at the included venues, as this could be a key part in any stewardship interventions. Any intervention needs to include both the supply and demand challenges and drivers, which is also clear from several of the challenges mentioned which related to the knowledge, attitudes and practices of customers, without having insight in their perceptions. The need for a holistic approach could be reflected upon in the discussion and future directions as part of the related limitations already mentioned.

Finally, even though the authors explained that the NYU Accra campus affirmed that IRB approval from NYU Abu Dhabi would be sufficient, it remains a serious concern that this potentially sensitive study did not request for IRB approval from Ghanian authorities. For most international research, IRB approval is sought in the country where data are collected, as well as in the country of origin of the main researchers. Are the Ghanian authorities aware?

Reviewer #4: • What are the main claims of the paper and how significant are they for the discipline?

The paper claims to investigate the challenges faced by medicine dispensers in Ghana from the perspective of the medicine vendors themselves. The topic was interesting as was the approach and the socio-ecological conceptual framework. The key challenges identified by the different types of medicine vendors included financial challenges faced by customers to pay the full cost of medicines, challenges in the treatment of illnesses like the vendors’ knowledge and skills and lack of diagnostics, and broader health system challenges of a fragmented health system. These findings are all very well known in the wider literature from Africa and from other parts of the world including South and South East Asia. I felt that there were several gaps in the analysis due to which the authors have not been able to distinctly and persuasively present the novelty of their findings or highlight how their novel approach has revealed a fresh set of insights. So, it’s difficult to justify that this paper makes a significant contribution to the existing literature on community based dispensing of medicines and healthcare.

• Are the claims properly placed in the context of the previous literature? Have the authors treated the literature fairly?

There are ample references to previous literature, more than a hundred in fact, which seems

rather excessive. Most are from Africa. I think the number of references could be reduced to about 60 and used more effectively and impactfully to make key points. It would also have been useful to bring in literature from a few countries like India and Bangladesh where there is a large community based informal sector that provides medicines including antibiotics at first contact. (See Gautham, M., et al., What are the challenges for antibiotic stewardship at the community level? An analysis of the drivers of antibiotic provision by informal healthcare providers in rural India. Social Science & Medicine, 2021. 275: p. 113813.)

Many of the results sections begin with references to key studies that have shown similar findings. This use of literature in the results section was unnecessary as the results section should focus only on presenting the key findings, not on introducing them or discussing them vis a vis other studies. And I feel some key aspects in the literature have been missed out while making recommendations – for example, rapid diagnostics for antimalarials has been found to lead to unnecessary use of antibiotics in Africa, implying that there could be unintended consequences of diagnostics.

Do the data and analyses fully support the claims? If not, what other evidence is required?

I think the data needs to be analysed in greater depth and more critically in order to show the novelty of the findings and how these build on the existing literature. The key limitations are:

-The differences between the trained pharmacists and the medicine vendors need to be brought out more clearly with respect to their profiles as well as their responses. Did they face similar challenges or were there major differences?

-There are quite a few overlaps in the way the results are organized. For example, lack of diagnostics appears in organizational level, community level and public policy level, (but is missing in Fig 1). One way of presenting the sub themes under each level more clearly might be to provide clear sub-headings under each level, and make sure you only include the points relevant to that sub-theme under that sub-heading. The results are rather rambling at present and it is very difficult to understand what point you are trying to make. Before you submit a revised version, please do ask a couple of colleagues/friends who have not been involved in this paper to do an independent review and get their feedback.

-You need to present the quotes in the appropriate style of a qualitative manuscript which is not the case at present. Only small quotes can be integrated into the text but the larger ones need to be presented separately, below the text, and the source needs to be added (anonymised but with an ID number and role). Please see the sample paper I have referenced earlier.

- You have skipped questions related to antibiotics for the non -pharmacists as they are not legally allowed to sell antibiotics. This is a HUGE limitation and must be acknowledged upfront because it severely limits what inferences you can draw about their antibiotic use. Perhaps you should limit this paper only to the qualified pharmacists.

-The discussion should not repeat the findings but explain and discuss their implications. The last three recommendations are disconnected with the key findings. Another concern I have is that you have no policy recommendations for such a fragmented healthcare scenario.

• If the paper is considered unsuitable for publication in its present form, does the study itself show sufficient potential that the authors should be encouraged to resubmit a revised version?

Yes, I think that with an improved analysis and significantly improved presentation of the findings, the authors should be encouraged to resubmit a revised version.

• Are details of the methodology sufficient to allow the experiments to be reproduced?

No. It is not clear who administered the survey, and what probing was done to elicit responses to the question on challenges. Was the conceptual framework used to shape your questions, or was it used only later to guide the analysis? What was the total number of shops and pharmacies that you mapped? Was it 89? Can you provide a distribution of these by type of shop.

• Is the manuscript well organized and written clearly enough to be accessible to non-specialists?

In general I found the introduction, methods and conceptual framework well organised and well written but the results and discussion need a lot more work to be clear and articulate for both specialist and non-specialist readers. Please see my specific comments in the attached word doc.

7. PLOS authors have the option to publish the peer review history of their article (what does this mean?). If published, this will include your full peer review and any attached files.

Reviewer #1: No

Reviewer #3: No

Reviewer #4: No

---

## [Author Response · Author response to Decision Letter 1]

10 Apr 2024

Responses to reviewer comments have been uploaded as a separate file.

---

## [Decision Letter · Decision Letter 2]

2 May 2024

Challenges in the distribution of antimicrobial medications in community dispensaries in Accra, Ghana

PONE-D-23-02556R2

Dear Dr. Frimpong,

We’re pleased to inform you that your manuscript has been judged scientifically suitable for publication and will be formally accepted for publication once it meets all outstanding technical requirements.

Kind regards,

Francesca Baratta, PharmD, PhD

Academic Editor

PLOS ONE

Reviewers' comments:

Reviewer's Responses to Questions

**Comments to the Author**

1. If the authors have adequately addressed your comments raised in a previous round of review and you feel that this manuscript is now acceptable for publication, you may indicate that here to bypass the “Comments to the Author” section, enter your conflict of interest statement in the “Confidential to Editor” section, and submit your "Accept" recommendation.

Reviewer #3: (No Response)

2. Is the manuscript technically sound, and do the data support the conclusions?

Reviewer #3: Partly

3. Has the statistical analysis been performed appropriately and rigorously? 

Reviewer #3: N/A

4. Have the authors made all data underlying the findings in their manuscript fully available?

Reviewer #3: Yes

5. Is the manuscript presented in an intelligible fashion and written in standard English?

Reviewer #3: Yes

6. Review Comments to the Author

Reviewer #3: Thanks for addressing most of the suggestions and concerns, which has improved the paper and readability.

I also agree the data are interesting, although the given methodological limitations (such as absence of a sampling phrame) and the significant potential of biased responses, continue to make it quite hard to draw any generalising conclusions. As such, it might be more of a pilot/feasibility study, than a study with results to be validated.

7. PLOS authors have the option to publish the peer review history of their article (what does this mean?). If published, this will include your full peer review and any attached files.

Reviewer #3: No

---

## [Editor Report · Acceptance letter]

16 May 2024

PONE-D-23-02556R2 

PLOS ONE

Dear Dr. Frimpong, 

I'm pleased to inform you that your manuscript has been deemed suitable for publication in PLOS ONE. Congratulations! Your manuscript is now being handed over to our production team.

Kind regards, 

on behalf of

Dr. Francesca Baratta 

Academic Editor

PLOS ONE